# ANALYZING PRIVACY LOSS IN UPDATES OF NATURAL LANGUAGE MODELS

## ABSTRACT

To continuously improve quality and reflect changes in data, machine learning-based services have to regularly re-train and update their core models. In the setting of language models, we show that a comparative analysis of model snapshots before and after an update can reveal a surprising amount of detailed information about the changes in the data used for training before and after the update. We discuss the privacy implications of our findings, propose mitigation strategies and evaluate their effect.

## 1 INTRODUCTION

Over the last few years, deep learning has made sufficient progress to be integrated into intelligent, user-facing systems, which means that machine learning models are now part of the regular software development lifecycle. As part of this move towards concrete products, models are regularly re-trained to improve performance when new (and more) data becomes available, to handle distributional shift as usage patterns change, and to respect user requests for removal of their data.

In this work, we show that model updates[1] reveal a surprising amount of information about changes in the training data, in part, caused by neural network's tendency to memorize input data. As a consequence, we can infer fine-grained information about differences in the training data by comparing two trained models even when the change to the data is as small as 0.0001% of the original dataset. This has severe implications for deploying machine learning models trained on user data, some of them counter-intuitive: for example, honoring a request to remove a user's data from the training corpus can mean that their data becomes exposed by releasing an updated model trained without it. This effect also needs to be considered when using public snapshots of high-capacity models (e.g. BERT (Devlin et al., 2019)) that are then fine-tuned on smaller, private datasets.

We study the privacy implications of *language model* updates, motivated by their frequent deployment on end-user systems (as opposed to cloud services): for instance, smartphones are routinely shipped with (simple) language models to power predictive keyboards. The privacy issues caused by the memorizing behavior of language models have recently been studied by Carlini et al. (2018), who showed that it is sometimes possible to extract out-of-distribution samples inserted into the training data of a model. In contrast, we focus on in-distribution data, but consider the case of having access to two versions of the model. A similar setting has recently been investigated by Salem et al. (2019a) with a focus on fully-connected and convolutional architectures applied to image classification, whereas we focus on natural language.

We first introduce our setting and methodology in Section 2, defining the notion of a *differential score* of token sequences with respect to two models. This score reflects the changes in the probabilities of individual tokens in a sequence. We then show how beam search can find token sequences with high differential score and thus recover information about differences in the training data. Our experiments in Section 3 show that our method works in practice on a number of datasets and model architectures including recurrent neural networks and modern transformer architectures. Specifically, we consider a) a synthetic worst-case scenario where the data used to train two model snapshots differs only in a canary phrase that was inserted multiple times; b) a more realistic scenario where we compare

---

[1]We use the term "model update" to refer to an update in the parameters of the model, caused for example by a training run on changed data. This is distinct from an update to the model architecture, which changes the number or use of parameters.

a model trained on Reddit comments with one that was trained on the same data augmented with subject-specific conversations. We show that an adversary who can query two model snapshots for predictions can recover the canary phrase in the former scenario, and fragments of discourse from conversations in the latter. Moreover, in order to learn information about such model updates, the adversary does not require any information about the data used for training of the models nor knowledge of model parameters or its architecture.

Finally, we discuss mitigations such as training with differential privacy in Section 4. While differential privacy grants some level of protection against our attacks, it incurs a substantial decrease in accuracy and a high computational cost.

## 2 METHODOLOGY

### 2.1 NOTATION

Let $T$ be a finite set of tokens, $T^*$ be the set of finite token sequences, and $Dist(T)$ denote the set of probability distributions over tokens. A language model $M$ is a function $M : T^* \to Dist(T)$, where $M(t_1 \ldots t_{i-1})(t_i)$ denotes the probability that the model assigns to token $t_i \in T$ after reading the sequence $t_1 \ldots t_{i-1} \in T^*$. We often write $M_D$ to make explicit that a multiset (i.e., a set that can contain multiple occurrences of each element) $D \subseteq T^*$ was used to train the language model.

### 2.2 ADVERSARY MODEL

We consider an adversary that has query access to two language models $M_D$, $M_{D'}$ that were trained on datasets $D, D'$ respectively (in the following, we use $M$ and $M'$ as shorthand for $M_D$ and $M_{D'}$). The adversary can query the models with any sequence $s \in T^*$ and observe the corresponding outputs $M_D(s), M_{D'}(s) \in Dist(T)$. The goal of the adversary is to infer information about the difference between the datasets $D, D'$.

This scenario corresponds to the case of language models deployed to client devices, for example in "smart" software keyboards or more advanced applications such as grammar correction.

### 2.3 DIFFERENTIAL RANK

Our goal is to identify the token sequences whose probability differs most between $M$ and $M'$, as these are most likely to be related to the differences between $D$ and $D'$.

To capture this notion formally, we define the *differential score DS* of token sequences, which is simply the sum of the differences of (contextualized) per-token probabilities. We also define a *relative* variant $\widetilde{DS}$ based on the relative change in probabilities, which we found to be more robust w.r.t. the "noise" introduced by different random initializations of the models $M$ and $M'$.

**Definition 1.** Given two language models $M, M'$ and a token sequence $t_1 \ldots t_n \in T^*$, we define the *differential score* of a token as the increase in its probability and the *relative differential score* as the relative increase in its probability. We lift these concepts to token sequences by defining

$$DS_M^{M'}(t_1 \ldots t_n) = \sum_{i=1}^{n} M'(t_1 \ldots t_{i-1})(t_i) - M(t_1 \ldots t_{i-1})(t_i),$$

$$\widetilde{DS}_M^{M'}(t_1 \ldots t_n) = \sum_{i=1}^{n} \frac{M'(t_1 \ldots t_{i-1})(t_i) - M(t_1 \ldots t_{i-1})(t_i)}{M(t_1 \ldots t_{i-1})(t_i)}.$$

The differential score of a token sequence is best interpreted relative to that of other token sequences. This motivates ranking sequences according to their differential score.

**Definition 2.** We define the *differential rank* $DR(s)$ of $s \in T^*$ as the number of token sequences of length $|s|$ with differential score higher than $s$.

$$DR(s) = \left| \left\{ s' \in T^{|s|} \, \middle| \, DS_M^{M'}(s') > DS_M^{M'}(s) \right\} \right|$$

The lower the rank of $s$, the more $s$ is exposed by a model update.

## 2.4 APPROXIMATING DIFFERENTIAL RANK

Our goal is to identify the token sequences that are most exposed by a model update, i.e., the sequences with the lowest differential rank (highest differential score). Exact computation of the differential rank for sequences of length $n$ requires exploring a search space of size $|T|^n$. To overcome this exponential blow-up, we propose a heuristic based on beam search.

At time step $i$, a beam search of width $k$ maintains a set of $k$ candidate sequences of length $i$. Beam search considers all possible $k|T|$ single token extensions of these sequences, computes their differential scores and keeps the $k$ highest-scoring sequences of length $i + 1$ among them for the next step. Eventually, the search completes and returns a set $S \subseteq T^n$.

We approximate the differential rank $DR(s)$ of a sequence $s$ by its rank among the sequences in the set $S$ computed by beam search, i.e. $\left| \{ s' \in S \mid DS_M^{M'}(s') > DS_M^{M'}(s) \} \right|$. The beam width $k$ governs a trade-off between computational cost and precision of the result. For a sufficiently large width, $S = T^{|s|}$ and the result is the true rank of $s$. For smaller beam widths, the result is a lower bound on $DR(s)$ as the search may miss sequences with higher differential score than those in $S$.

In experiments, we found that *shrinking* the beam width as the search progresses speeds the search considerably without compromising on the quality of results. Initially, we use a beam width $|T|$, which we half at each iteration (i.e., we consider $|T|/2$ candidate phrases of length two, $|T|/4$ sequences of length three, ...).

## 3 EXPERIMENTAL RESULTS

In this section we report on experiments in which we evaluate privacy in language model updates using the methodology described in Section 2. We begin by describing the experimental setup.

## 3.1 SETUP

For our experiments, we consider three datasets of different size and complexity, matched with standard baseline model architectures whose capacity we adapted to the data size. All of our models are implemented in TensorFlow. Note that the random seeds of the models are *not* fixed, so repeated training runs of a model on an unchanged dataset will yield (slightly) different results. We will release the source code as well as analysis tools used in our experimental evaluation at `https://double/blind`.

Concretely, we use the Penn Treebank (Marcus et al., 1993) (PTB) dataset as a representative of low-data scenarios, as the standard training dataset has only around 900 000 tokens and a vocabulary size of 10 000. As corresponding model, we use a two-layer recurrent neural network using LSTM cells with 200-dimensional embeddings and hidden states and no additional regularization (this corresponds to the *small* configuration of Zaremba et al. (2014)).

Second, we use a dataset of Reddit comments with 20 million tokens overall, of which we split off 5% as validation set. We use a vocabulary size of 10 000. As corresponding model, we rely on a one-layer recurrent neural network using an LSTM cell with 512-dimensional hidden states and 160-dimensional embeddings, using dropout on inputs and outputs with a keep rate of 0.9 as regularizer. These parameters were chosen in line with a neural language model suitable for next-word recommendations on resource-bounded mobile devices. We additionally consider a model based on the Transformer architecture (Vaswani et al., 2017) (more concretely, using the BERT (Devlin et al., 2019) codebase) with four layers of six attention heads each with a hidden dimension of 192.

Finally, we use the Wikitext-103 dataset (Merity et al., 2017) with 103 million training tokens as a representative of a big data regime, using a vocabulary size of 20 000. As model, we employ a two-layer RNN with 512-dimensional LSTM cells and token embedding size 512 and again dropout on inputs and outputs with a keep rate of 0.9 as regularizer. We combined this large dataset with this relatively low-capacity model (at least according to the standards of the state of the art in language modeling) to test if our analysis results still hold on datasets that clearly require more model capacity than is available.

Table 1: Differential score ($DS$) for PTB, Reddit and Wikitext-103 dataset across different token and phrase frequencies. White cell background means that the differential rank $DR$ (as approximated by our beam search) of the phrase is 0, grey cell background means that $DR$ is >1000.

| Dataset | Penn Treebank | | | Reddit | | | | | | Wikitext-103 | |
|---|---|---|---|---|---|---|---|---|---|---|---|
| Model Type (Perplexity) | RNN (120.90) | | | RNN (79.63) | | | Transformer (69.29) | | | RNN (48.59) | |
| Canary Token Freq. | 1:18K | 1:3.6K | 1:1.8K | 1:1M | 1:100K | 1:10K | 1:1M | 1:100K | 1:10K | 1:1M | 1:200K |
| All Low | 3.40 | 3.94 | 3.97 | 2.83 | 3.91 | 3.96 | 3.22 | 3.97 | 3.99 | 1.39 | 3.81 |
| Low to High | 3.52 | 3.85 | 3.97 | 0.42 | 3.66 | 3.98 | 0.25 | 3.66 | 3.97 | 0.07 | 3.21 |
| Mixed | 3.02 | 3.61 | 3.90 | 0.23 | 3.04 | 3.92 | 0.39 | 3.25 | 3.96 | 0.25 | 3.02 |
| High to Low | 1.96 | 2.83 | 3.46 | 0.74 | 1.59 | 2.89 | 0.18 | 1.87 | 3.10 | 0.08 | 1.22 |

## 3.2 PRIVACY ANALYSIS OF MODEL UPDATES USING SYNTHETIC CANARIES

We first study the privacy implications of model updates in controlled experiments with synthetic data. To this end, we create a number of canary phrases that serve as a proxy for private data (they are grammatically correct and do not appear in the original dataset) and that exhibit a variety of different token frequency characteristics.

Specifically, we fix the length of the canary phrase to 5, choose a valid phrase structure (e.g. Subject, Verb, Adverb, Compound Object), and instantiate each placeholder with a token that has the desired frequency in $D$. We create canaries in which frequencies of tokens are *all low* (all tokens are from the least frequent quintile of words), *mixed* (one token from each quintile), *increasing from low to high*, and *decreasing from high to low*. As the vocabularies differ between the different datasets, the canaries are dataset-dependent. For example, the *mixed* phrase across all the datasets is "NASA used deadly carbon devices," and the *all low* phrase for PTB is "nurses nervously trusted incompetent graduates."

For a given dataset $D$ and a canary phrase $s \notin D$, we construct a dataset $D_{+k*s}$ by inserting $k$ copies of $s$ into $D$. We use the differential score $DS$ and the differential rank $DR$ of canary phrase $s$ to answer a number of research questions on our model/dataset combinations. Note that analyzing *removal* of specific phrases from the dataset simply requires swapping the role of $D$ and $D_{+k*s}$.

**RQ1: What is the effect of the number of canary phrase insertions?** We consider different numbers of insertions, adapted to the number of tokens in the training corpus:[2]

- For PTB, we consider $k \in \{10, 50, 100\}$ canary insertions (corresponding to 1 canary token in 18K training tokens, 1 in 3.6K, and 1 in 1.8K).

- For the Reddit dataset, we use $k \in \{5, 50, 500\}$ (corresponding to 1 in 1M, 1 in 100K, 1 in 10K).

- For the Wikitext-103 data, we use $k \in \{20, 100\}$ (corresponding to 1 in 1M, 1 in 200K).

Table 1 summarizes all of our experiments. As expected, the differential score of canaries grows monotonously with the number of insertions, for all kinds of canaries and models. More surprisingly, in cells with white background, the canary phrase has the *maximum differential score* among all token sequences found by our beam search, i.e. it ranks first. This means that the canary phrase can easily be extracted without any prior knowledge about it or the context in which it appears (this is in contrast to the single-model results of Carlini et al. (2018), who assumed a known prefix). The signal for extraction is strong even when the inserted canaries account for only 0.0001% of the tokens in the dataset. This becomes visible in the first row of Table 1 where differential scores approaches 4, which is close to the upper bound of 5 (for 5-token canaries).

**RQ2: What is the effect of token frequency in training data?** Comparing the columns of Table 1 can be used to answer this question:

- Phrases with all low-frequency tokens consistently show the highest differential score. Such phrases rank first even when the model is updated with the smallest number of canary

---

[2]The non-aligned insertion frequencies are due to legacy reasons in our experiments. For the final version we will re-run all experiments with aligned frequencies.

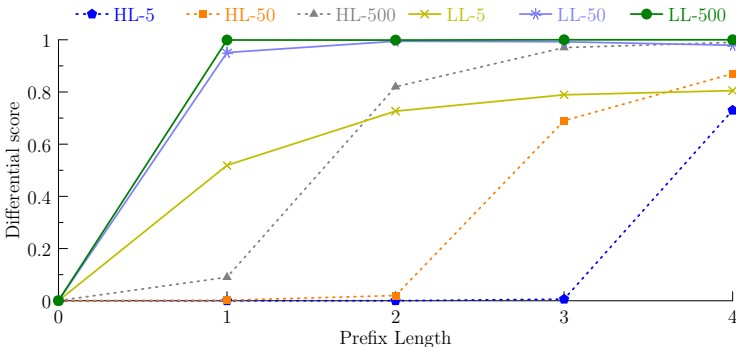

Figure 1: Differential score of tokens in canaries given a prefix for the Reddit dataset. Dashed (solid) lines represent experiments with $k$ insertions of canaries with all-low (resp. high-to-low) token frequencies, indicated by LL-$k$ (resp. HL-$k$).

insertions, as seen in the first row of Table 1. This means that phrases composed of rare words are more likely to be exposed in model updates than other phrases.

- Canary phrases that start with a low-frequency token, followed by tokens of increasing or mixed frequency, have higher rank than canaries with all low-frequency tokens, but become exposed for a moderate number of insertions into the dataset, see rows 2 and 3 of Table 1.

- Canaries composed of tokens with descending frequency are the least susceptible to our analysis and tolerate a higher number of insertions before they become exposed. This is expected, as our beam search is biased towards finding high-scoring prefixes.

**RQ3: What is the effect of knowledge about the canary context?**   We evaluate the differential score of suffixes of our canary phrases assuming knowledge of a prefix. This gives insight into the extent to which an attacker with background knowledge can extract a secret. To this end we consider a dataset $D$, a canary phrase $s = t_1 \ldots t_n \notin D$ and the augmented dataset $D_{+k*s}$. For $i = 1, \ldots, n$ we take the prefix $t_1 \ldots t_{i-1}$ of the canary phrase and compute the differential score $r$ of the token $t_i$ conditional on having read the prefix, i.e. $M'(t_1 \ldots t_{i-1})(t_i) - M(t_1 \ldots t_{i-1})(t_i)$. The relationship between $i$ and $r$ indicates how much knowledge about $s$ is required to expose the remainder of the canary phrase.

Figure 1 depicts the result of this analysis for canaries with high-to-low and all-low token frequencies on the Reddit dataset. Our results show that, while the differential score of the first token without context is close to zero, the score of subsequent tokens quickly grows for all-low canaries, even with a low number of canary insertions. In contrast, more context is required before observing a change in the differential score of high-to-low canaries, as the model is less influenced by the small number of additional occurrences of frequent tokens. This suggests that, even in cases where we fail to extract the canary without additional knowledge (see **RQ1** above), an adversary can still use the differential rank to complete a partially known phrase, or confirm that a canary phrase was used to update the dataset.

**RQ3A: What is the effect of inserting canaries together with additional data?**   We consider the setting where the model is re-trained on data consisting of the canary phrase and some fresh in-distribution data $D_{extra}$ along with the original dataset $D_{orig}$. Concretely, we first split our Reddit dataset into $D_{orig}$ and $D_{extra}$ (such that the latter is $20\%, 50\%, 100\%$ of the size of $D_{orig}$). Then, we trained a model $M$ on reduced $D_{orig}$ and a second model $M'$ on $D = D_{orig} \cup D_{extra}$ and the inserted canaries. The results of this experiment are displayed in Table 2, where we can see that $DS_M^{M'}$ does not change significantly. Note that the $0\%$ column is identical to the result from Table 1. In conclusion, canaries can be extracted from the trained model even when they are contained in a substantial larger dataset extension.

**RQ3B: What is the effect of updating the model using a continued training approach?**   We also consider the setting of continued training, in which an existing model is trained on a new dataset

Table 2: $DS_M^{M'}$ of the mixed frequence canary phrase for the Reddit (RNN) model using different update techniques. We use $T(R, D_{orig}) \rightarrow M$ to denote that model $M$ was obtained by training on data $D_{orig}$ starting from model $R$. Here, $R$ are (fresh) random initial parameters, $D_{extra}$ is an additional dataset from the same distribution as $D_{orig}$, and $C$ is the set of canaries. Re-training refers to training on the data from scratch with a new random initialization, whereas continued training fine-tunes an existing model on fresh data. A white cell background means that the differential rank $DR$ (as approximated by our beam search) of the phrase is 0, grey cell background means that $DR$ is >1000.

| Training Method | Re-training $T(R, D_{orig} \cup D_{extra} \cup C) \rightarrow M'$ | | | | Continued Training 1 $T(M, D_{extra} \cup C) \rightarrow M'$ | | | Continued Training 2 $T(M, D_{extra} \cup C) \rightarrow \tilde{M}$ $T(\tilde{M}, D'_{extra}) \rightarrow M'$ |
|---|---|---|---|---|---|---|---|---|
| $|D_{extra}|/|D_{orig}|$ | 0% | 20% | 50% | 100% | 20% | 50% | 100% | 100% |
| 1: 1M | 0.23 | 0.224 | 0.223 | 0.229 | 0.52 | 0.34 | 0.46 | 0.01 |
| 1: 100K | 3.04 | 3.032 | 3.031 | 3.038 | 3.56 | 3.25 | 3.27 | 0.26 |

(e.g., pre-trained on a generic corpus and fine-tuned on a more specific dataset). For this, we train a model $M$ on a dataset $D_{orig}$ to convergence, and then continue training using the union of $D_{extra}$ and the canaries. We use the same dataset splits as in **RQ3A** to create the $D_{orig}$ and $D_{extra}$ datasets. The results of this experiment are shown in the middle column of Table 2. We observe that in all cases the differential score is higher for continued training than re-training. As expected, the differential score of the canary phrase decreases as additional extra data is used for fine-tuning. This shows that the risk of leaking canary phrases increases when a model is updated using the continued training approach.

Finally, we also consider a possible mitigation strategy, in which we perform continued training in two stages. For this, we split the dataset into three equal parts $D_{orig}$, $D_{extra}$ and $D'_{extra}$. We proceed as in the continued training setting above, but add a final step in which we train on another dataset after training on the canaries. This resembles a setting where an attacker does not have access to two consecutive snapshots. The results are on the right column of Table 2, showing that the differential score of the canary phrase drops substantially after the second training stage. Thus, two or multi-stage continued training might mitigate leakage of private data.

## 3.3 PRIVACY ANALYSIS OF MODEL UPDATES USING SUBJECT-SPECIFIC CONVERSATIONS

We now study the privacy implications of model updates in real-world scenarios. As a representative scenario, we compare models trained on the Reddit dataset against models trained on the same data augmented with messages from one of two newsgroups from the 20 Newsgroups dataset (Lang, 1995): a) `rec.sport.hockey`, containing around 184K tokens, approximately 1% of the original training data; and b) `talk.politics.mideast`, containing around 430K tokens, approximately 2% of the original training data. For both newsgroups, the number of tokens we insert is significantly larger than in the synthetic experiments of Section 3.2. However, we insert full conversations, many of which are of a general nature and off the topic of the newsgroup.

**RQ4: Do the results on synthetic data extend to real-world data?**  As opposed to synthetic experiments using canaries, when using real-world data there is no individual token sequence whose rank would serve as a clear indicator of a successful attack. We hence resort to a *qualitative* evaluation where we inspect the highest-scoring sequences found by beam search. Since the sequences returned by vanilla beam search typically share a common prefix, we alternatively run a *group beam search* to get a more representative sample: we split the initial $|T|$ one-token sequences into $N$ groups according to their differential score, and run parallel beam searches extending each of the groups independently.

Table 3 displays the result of our evaluation on Reddit augmented with `rec.sport.hockey`, i.e., the highest-scoring sequences of length 4 in each group of a $\widetilde{DS}$-based group beam search with $N = 5$ (top row) and $N = 1$ (bottom row). The exposed sentences are on-topic w.r.t. the newsgroup added, which suggests that the phrases with highest relative differential score are specific to the

newsgroup used and that, indeed, data used for the update is exposed. We obtain results of comparable relevance using the `talk.politics.mideast` newsgroup, which we report in Table 4.

Table 3: Top ranked phrases in vanilla and group beam search for RNN and Transformer models trained with `rec.sport.hockey`. For the layperson: NHL stands for National Hockey League; Los Angeles Kings, Minnesota North Stars, and Toronto Maple Leaf are NHL teams; Norm Green was the owner of the North Stars; an ice hockey game consists of three periods with overtime to break ties. Capitalization added for emphasis.

|  | RNN | | Transformer | |
|---|---|---|---|---|
|  | Phrase | $\widetilde{DS}$ | Phrase | $\widetilde{DS}$ |
| Group | Angeles Kings prize pools | 56.42 | Minnesota North Stars playoff | 96.81 |
|  | National Hockey League champions | 53.68 | Arsenal Maple Leaf fans | 71.88 |
|  | Norm 's advocate is | 39.66 | Overtime no scoring chance | 54.77 |
|  | Intention you lecture me | 21.59 | Period 2 power play | 47,85 |
|  | Covering yourself basically means | 21.41 | Penalty shot playoff results | 42.63 |
| Vanilla | National Hockey League champions | 53.68 | Minnesota North Stars playoff | 96.81 |
|  | National Hockey League hockey | 52.82 | Minnesota North Stars 3 | 96.78 |
|  | National Hockey League cup | 52.44 | Minnesota North Stars hockey | 96.76 |
|  | National Hockey League USA | 52.34 | Minnesota North Stars games | 96.75 |
|  | National Hockey League playoff | 52.18 | Minnesota North Stars 5 | 96.72 |

Table 4: Top ranked phrases in vanilla and group beam search for RNN and Transformer models trained with `talk.politics.mideast`. Center for Policy Research is a prolific newsgroup poster; Professor of History is a common title appearing in message signatures; many of the posts around the time the 20 Newsgroups dataset (Lang, 1995) was collected discuss tensions between Turkey and Armenia.

|  | RNN | | Transformer | |
|---|---|---|---|---|
|  | Phrase | $\widetilde{DS}$ | Phrase | $\widetilde{DS}$ |
| Group | Turkey searched first aid | 31.32 | Center for Policy Research | 200.27 |
|  | Doll flies lay scattered | 22.79 | Escaped of course ... | 95.18 |
|  | Arab governments invaded Turkey | 20.20 | Holocaust %UNK% museum museum | 88.20 |
|  | Lawsuit offers crime rates | 18.35 | Troops surrounded village after | 79.35 |
|  | Sanity boosters health care | 11.17 | Turkey searched neither Arab | 37.69 |
| Vanilla | Turkey searched first aid | 31.32 | Center for Policy Research | 200.27 |
|  | Turkey searched a plane | 24.63 | Professor of History I | 120.98 |
|  | Turkey searched supreme soviet | 21.59 | Professor of History History | 120.93 |
|  | Turkey searched arab Turkey | 20.34 | Professor of History we | 120.92 |
|  | Turkey searched national plane | 19.44 | Professor of History he | 120.92 |

**RQ4A: What is the effect of re-training with additional public data?** Similarly to RQ3A in Section 3.2, we consider partitions of the Reddit dataset into datasets $D_{orig}$ and $D_{extra}$ of different relative sizes. For each partition, we compare a model $M$ trained on $D_{orig}$ to a model $M'$ trained on $D_{extra} \cup N$, where $N$ are all messages from one newsgroup from the 20 Newsgroups dataset. We sample a few representative phrases from group beam searches on all pairs of models and compare their relative differential scores.

In all cases, a group beam search can recover subject-specific phrases from newsgroups discussions. Some of the phrases resemble canaries, in that they occur multiple times literally in the datasets (e.g. `Center for Policy Research`), while others never occur literally but digest recurrent discussions (e.g. `Partition of northern Israel`). Table 5 shows the relative differential score of these phrases for different partitions. As observed for canaries, scores vary little when additional data is used during re-training.

**RQ4B: What is the effect of continued training?** Using the same dataset splits as in RQ4A and similarly to RQ3B in Section 3.2, we compare a model $M$ trained from scratch on $D_{orig}$ to a model $M'$ obtained by continuing training $M$ on $D_{extra} \cup N$. The results are shown in Table 5. The last two rows contain phrases found by group beam search on $M$ and a model $M'$ obtained from $M$ by continued training, but that have too low a score to be found when $M'$ is re-trained

from scratch instead. The converse, i.e. phrases that have low score when continuing training and high score when re-training, seems to occur rarely and less consistently (e.g. `Saudi troops surrounded village`). For canary-like phrases, the results are in line with those in Table 2, with scores decreasing as more data is used during the fine-tuning stage. For other phrases, the results are not as clear-cut. While fine-tuning a model exclusively on private data yields scores that are significantly higher than when re-training a model from scratch, this effect vanishes as more additional data is used; in some cases continued training yields scores lower than when re-training a model on the same data.

Table 5: Relative differential score of phrases found by beam search when re-training from scratch and continuing training from a previous model. The results are for RNN models trained on partitions of the Reddit dataset with $N = $ `talk.politics.mideast`. Cells for which continued training yields a higher score than re-training appear in bold font. Capitalization added for emphasis.

| Phrase | Re-training $T(R, D_{orig} \cup D_{extra} \cup N) \to M'$ | | | | | Continued Training $T(M, D_{extra} \cup N) \to M'$ | | | | |
|---|---|---|---|---|---|---|---|---|---|---|
| $\|D_{extra}\|/\|D_{orig}\|$ | 0% | 5% | 10% | 25% | 100% | 0% | 5% | 10% | 25% | 100% |
| Center for Policy Research | 99.77 | 101.38 | 97.11 | 98.65 | 91.53 | **276.98** | **198.69** | **150.56** | **122.25** | **117.54** |
| Troops surrounded village after | 44.50 | 44.50 | 44.50 | 44.41 | 44.54 | **173.95** | **47.38** | 19.48 | 7.81 | 35.56 |
| Partition of northern Israel | 27.61 | 16.81 | 38.48 | 26.10 | 38.76 | **68.98** | 16.48 | 12.47 | 22.93 | 18.82 |
| West Bank peace talks | 25.68 | 25.64 | 25.69 | 25.71 | 25.75 | **71.54** | 24.38 | **28.60** | 16.91 | 4.62 |
| Spiritual and political leaders | 25.23 | 25.98 | 17.04 | 24.21 | 23.47 | **126.92** | 14.91 | 10.00 | 3.44 | 11.05 |
| Saudi troops surrounded village | 24.31 | 24.31 | 24.31 | 24.31 | 24.30 | 5.05 | **44.58** | 7.29 | **63.84** | |
| Arab governments invaded Turkey | 22.59 | 22.62 | 22.80 | 22.78 | 22.80 | **24.01** | 15.58 | 7.08 | 18.12 | 11.90 |
| Little resistance was offered | 22.24 | 22.09 | 25.12 | 22.34 | 25.59 | **215.16** | **25.02** | 2.00 | 3.30 | 5.64 |
| Buffer zone aimed at protecting | 4.00 | 4.47 | 5.30 | 5.25 | 5.69 | **57.29** | **69.76** | **18.92** | **14.50** | **22.25** |
| Capital letters racial discrimination | 3.76 | 3.32 | 3.40 | 3.60 | 3.84 | **94.60** | **52.74** | **39.11** | **11.22** | 3.45 |

## 4 MITIGATION VIA DIFFERENTIAL PRIVACY

In this section we explore how differential privacy can be used to mitigate the information leakage induced by a model update. Differential privacy (DP) (Dwork & Roth, 2014) provides strong guarantees on the amount of information leaked by a released output. Given a computation over records it guarantees limits on the the effect that any input record can have on the output. Formally, $F$ is a $(\epsilon, \delta)$-differentially-private computation if for any datasets $D$ and $D'$ that differ in one record and for any subset $O$ of possible outputs of $F$ we have

$$\Pr(F(D) \in O) \leq \exp(\epsilon) \cdot \Pr(F(D') \in O) + \delta.$$

At a high level, differential privacy can be enforced in gradient-based optimization computations (Abadi et al., 2016; Song et al., 2013; Bassily et al., 2014) by clipping the gradient of every record in a batch according to some bound $L$, then adding noise proportional to $L$ to the sum of the clipped gradients, averaging over the batch size and using this noisy average gradient update during backpropagation.

Differential privacy is a natural candidate for defending against membership-like inferences about the input data. The exact application of differential privacy for protecting the information in the model update depends on what one wishes to protect w.r.t. the new data: individual sentences in the new data or all information present in the update. For the former, sequence-level privacy can suffice while for the latter group DP can serve as a mitigation technique where the size of the group is proportional to the number of sequences in the update. Recall that an $\epsilon$-DP algorithm $F$ is $k\epsilon$ differentially private for groups of size $k$ (Dwork & Roth, 2014).

**RQ5: Does DP protect against phrase extraction based on differential ranks?** We evaluate the extent to which DP mitigates attacks considered in this paper by training models on the Penn Treebank dataset with canaries with sequence-level differential privacy. We train DP models using the TensorFlow Privacy library (Andrew et al., 2019) for two sets of $(\epsilon, \delta)$ parameters $(5, 1 \times 10^{-5})$ and $(111, 1 \times 10^{-5})$ for two datasets: PTB and PTB with 50 insertions of the all-low-frequency canary. We rely on (Andrew et al., 2019) to train models with differentially private stochastic gradient descent using Gaussian noise mechanism and to compute the overall privacy loss of the training phase.

As expected, the performance of models trained with DP degrades, in our case from $\approx$23% accuracy in predicting the next token on the validation dataset to 11.89% and 13.34% for $\epsilon$ values of 5 and

111, respectively. While the beam search with the parameters used in Section 3.2 does not return the canary phrase for the DP-trained models anymore, we note that the models have degraded so far that they are essentially only predicting the most common words from each class ("is" when a verb is required, ...) and thus, the result is unsurprising. We note that the guarantees of sequence-level DP formally do not apply for the case where canary phrases are inserted as multiple sequences, and that $\epsilon$ values for our models are high. However, the $\epsilon$-analysis is an upper bound and similar observations about the effectiveness of training with DP with high $\epsilon$ were reported by Carlini et al. (2018).

We further investigate the effect of DP training on the differential rank of a canary phrase that was inserted 50 times. Instead of using our beam search method to approximate the differential rank, we fully explore the space of subsequences of length two, and find that the $DR$ for the two-token prefix of our canary phrase dropped from 0 to $9\,458\,399$ and $849\,685$ for the models with $\epsilon = 5$ and $\epsilon = 111$ respectively. In addition, we compare the differential score of the whole phrase and observe that it drops from 3.94 for the original model to $4.5 \times 10^{-4}$ and $2.1 \times 10^{-3}$ for models with $\epsilon = 5$ and $\epsilon = 111$ respectively.

Though our experiment results validate that DP can mitigate the particular attack method considered in this paper for canary phrases, the model degradation is significant. In addition, the computational overhead of per-sequence gradient clipping is substantial, making it unsuitable for training high-capacity neural language models on large datasets.

## 5 RELATED WORK

In recent years several works have identified how machine learning models leak information about private training data. Membership attacks introduced by Shokri et al. (2017) show that one can identify whether a record belongs to the training dataset of a classification model given black-box access to the model and shadow models trained on data from a similar distribution. Salem et al. (2019b) demonstrate that similar attacks are effective under weaker adversary models. Attribute inference attacks (Fredrikson et al., 2015), which leak the value of sensitive attributes of training records, have been shown successful for regression and classification models. In the distributed learning setting, Hitaj et al. (2017) and Melis et al. (2018) demonstrate that individual gradient updates to a model can reveal features specific to one's private dataset.

Carlini et al. (2018) is closest to our work, as it also considers information leakage of language models. The authors assess the risk of (unintended) memorization of rare sequences in the training data by introducing an *exposure* metric. They show that exposure values can be used to retrieve canaries inserted into training data from a character-level language model. The key differences to our approach are that 1) we consider an adversary that has access to two snapshots of a model, and 2) our canaries are grammatically correct sentences (i.e., follow the distribution of the data) whereas Carlini et al. (2018) add a random sequence of numbers in a fixed context (e.g., "The random number is ...") into a dataset of financial news articles, where such phrases are rare. We instead consider the scenario of extracting canaries *without any context*, even if the canary token frequency in the training dataset is as low as one in a million, and for canary phrases that are more similar to the training data.

Song & Shmatikov (2018) also study sequence-to-sequence language models and show how a user can check if their data has been used for training. In their setting, an auditor needs an auxiliary dataset to train shadow models with the same algorithm as the target model and queries the target model for predictions on a sample of the user's data. The auxiliary dataset does not need to be drawn from the same distribution as the original training data (unlike Shokri et al. (2017)) and the auditor only observes a list of several top-ranked tokens. In contrast, our approach requires *no* auxiliary dataset, but assumes access to the probability distributions over all tokens from two different model snapshots. From this, we are able to recover full sequences from the differences in training data rather than binary information about data presence. Like them, we find that sequences with infrequent tokens provide a stronger signal to the adversary/auditor.

Salem et al. (2019a) consider reconstruction of training data that was used to update a model. While their goal is similar to ours, their adversarial model and setup differ: 1) similar to Song & Shmatikov (2018); Shokri et al. (2017) their attacker uses shadow models trained on auxiliary data drawn from the same distribution as the target training dataset, while in our setting the attacker has no prior knowledge of this distribution and does not need auxiliary data; 2) the updated model is obtained by

fine-tuning the target model with additional data rather than re-training it from scratch on the changed dataset; 3) the focus is on classification models and not on (generative) language models.

Information leakage from updates has also been considered in the setting of searchable encryption. An attacker who has control over data in an update to an encrypted database can learn information about the content of the database as well as previous encrypted searches on it (Cash et al., 2015). Pan-privacy (Dwork et al., 2010), on the other hand, studies the problem of maintaining differential privacy when an attacker observes snapshots of the internal state of a differentially-private algorithm between data updates.

In terms of defenses, McMahan et al. (2018) study how to train LSTM models with differential privacy guarantees at a user-level. They investigate utility and privacy trade-offs of the trained models depending on a range of parameters (e.g., clipping bound and batch size). Carlini et al. (2018) show that differential privacy protects against leakage of canaries in character-level models, while Song & Shmatikov (2018) show that an audit as described above fails when training language models with user-level differential privacy using the techniques of McMahan et al. (2018). Ginart et al. (2019) define deletion of a training data point from a model as an stochastic operation returning the same distribution as re-training from scratch without that point, and develop deletion algorithms for $k$-means clustering with low amortized cost. Publishing snapshots of a model before and after a deletion matches our adversarial model and our results apply.

## 6 CONCLUSION

As far as we know, this article presents the first systematic study of the privacy implications of releasing snapshots of a language model trained on overlapping data. We believe this is a realistic threat that needs to be considered in the lifecycle of machine learning applications. We aim to encourage the research community to work towards quantifying and reducing the exposure caused by model updates, and hope to make practitioners aware of the privacy implications of deploying high-capacity language models as well as their updates.

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
