# OpenReview forum: "Analyzing Privacy Loss in Updates of Natural Language Models"
_ICLR.cc/2020/Conference — Reject_

### Official Review · AnonReviewer1 · 2019-10-22
**Official Blind Review #1**

**Rating:** 6

**Review:**

Summary: This paper looks at privacy concerns regarding data for a specific model before and after a single update. It discusses the privacy concerns thoroughly and look at language modeling as a representative task. They find that there are plenty of cases namely when the composition of the sequences involve low frequency words, that a lot of information leak occurs.

Positives: The ideas and style of research is nice. This is an important problem and I think this paper does a good job investigating this in the context of language modeling. I do hope the community (and I think the community is) moving towards being aware of these sorts of privacy issues.

Concerns: I don't know how generalizable these results would be on really well-trained language models (rnn, convolution-based, or transformer-based). The related work section doesn't seem particularly well put together, so its difficult to place the work in appropriate context and gauge its impact.

Other Thoughts: I'd like more thorough error analysis looking at exactly what kinds of strings/more nuanced properties of sequences that get a high differential score.

Overall I think this work is interesting and I would encourage the authors to try and add as much quantitative evaluation as possible, but also try and include qualitative information regarding specific sequences after prodding the models. Those could go a long way in strengthening the paper.

**Experience Assessment:**

I have published one or two papers in this area.

**Review Assessment: Checking Correctness Of Derivations And Theory:**

I assessed the sensibility of the derivations and theory.

**Review Assessment: Checking Correctness Of Experiments:**

I assessed the sensibility of the experiments.

**Review Assessment: Thoroughness In Paper Reading:**

I read the paper at least twice and used my best judgement in assessing the paper.

---

> ### Author Response · Authors · 2019-11-08
> **Discussion of Review #1**
>
> Thank you for recognizing the importance of the problem and giving feedback to our submission.
>
> > Concerns: I don't know how generalizable these results would be on
> > really well-trained language models (rnn, convolution-based, or
> > transformer-based).
>
> It would be helpful if you could be more precise about what you mean by “really well-trained" here. The tested models are implementing standard practices of token-level language modeling (i.e., regularization via (recurrent) dropout, tying of token embeddings and output projection, …) but are not aiming to compete with recent high-capacity efforts such as GPT-2. The experiments show that our findings hold for both RNN-based and Transformer-based models (and it would be surprising if convnets would behave differently) of small and medium capacity. It seems intuitive that higher-capacity models would only exacerbate the problem, as they are known to be prone to memorizing substantial input chunks.
>
> Our paper tries to provide experimental evidence of an angle of attack that has not been studied well, and thus serves as a warning signal to practitioners that are deploying language models in the wild.
>
> > The related work section doesn't seem particularly well put together,
> > so its difficult to place the work in appropriate context and gauge
> > its impact.
>
> Our work falls under “attacks on ML models” topic and specifically language models. It is not yet well-understood what language models memorize and can leak; to this end, our paper proposes a new angle of extracting data that ML community has to be aware of. We believe we discuss the most relevant previous works in this topic, but we welcome suggestions about anything we may have missed.
>
> > Other Thoughts: I'd like more thorough error analysis looking at exactly
> > what kinds of strings/more nuanced properties of sequences that get
> > a high differential score.
>
> In initial experiments, we found that out-of-distribution sequences (such as the “My social security number is […]” used in Carlini et al. (2019)) are very easy to extract from the difference between two models. This is why we focused on manually creating sentences that are "near" to the training data by following a simple valid grammatical format and then varied the frequency of the used terms. In practice, it turns out that the models behave very much like expected: If canaries use very frequent words, extracting them becomes harder; if they use infrequent words, extracting them becomes easier.
>
> > Overall I think this work is interesting and I would encourage the
> > authors to try and add as much quantitative evaluation as possible,
> > but also try and include qualitative information regarding specific
> > sequences after prodding the models. Those could go a long way in
> > strengthening the paper.
>
> We have updated the paper with experiments using several new settings of data splits and continued training setup (for example,  where the model is trained on the original data, and then “fine-tuned” on a smaller additional dataset). We refer the reviewer for detailed explanation in the general response. New content is presented in Section 3.2 (RQ3A, RQ3B) of the updated submission.

---

> > ### Comment · AnonReviewer1 · 2019-11-15
> > **Reply to Author**
> >
> > By well-trained, I mean models that are empirically competitive on LM benchmarks. The point you make about larger/higher-capacity models memorizing large amounts of data is true and I'd also theorize that these models will exacerbate the problem that you observe. I didn't consider that initially.
> >
> > I appreciate the extra experiments that help clarify the approach. Thanks for clarifying! Overall I think the paper is still a weak accept and I recommend my initial score of a 6.

---

### Official Review · AnonReviewer3 · 2019-10-22
**Official Blind Review #3**

**Rating:** 3

**Review:**

This paper provides an empirical evaluation of the privacy implications of releasing updated versions of language models. The authors show how access to two sequential snapshots of a trained language model can reveal highly specific information about the content of the data used to update the model, even when that data is in-distribution.

The paper contains easy to understand, concrete experiments and results, but seems altogether a little underdeveloped. The methodology is sound, but the synthetic experiments around which much of the paper is based may not be sufficiently novel and give little indication of broader implications. It would have been more convincing if these results replicated with two splits of the same dataset, rather than identical datasets with one augmented by canary tokens.

The qualitative evaluation of subject-specific updates is also not sufficiently informative. It would have been useful to define a specific attack and see under what circumstances such an attack would succeed. In the current results, I am not convinced that any of the phrases in Table 3 represent a privacy violation.

The differential privacy experiment seems to be missing many details: what dataset was this trained on? Are the accuracy values for the training set or a separate testing set? Other works have shown that it is possible to train a differentially private language model without large sacrifices in accuracy, so it would be helpful to know what differentiates this experiment.

I would also note that the motivation, a predictive keyboard, is not a situation in which maximizing accuracy is generally desirable: users tend to find this creepy rather than helpful.

This is a nice idea but would benefit from some more polishing and more extensive testing.

**Experience Assessment:**

I do not know much about this area.

**Review Assessment: Checking Correctness Of Derivations And Theory:**

N/A

**Review Assessment: Checking Correctness Of Experiments:**

I assessed the sensibility of the experiments.

**Review Assessment: Thoroughness In Paper Reading:**

I read the paper at least twice and used my best judgement in assessing the paper.

---

> ### Author Response · Authors · 2019-11-08
> **Discussion of Review #3 (Part 1/2)**
>
> Thank you for your feedback and we hope that our comments below can resolve your concerns. Note that this response is split into two parts for character limit reasons.
>
> > the synthetic experiments around which much of the paper is based may
> > not be sufficiently novel
>
> The paper presents the first study of privacy implications of releasing snapshots of language models trained on overlapping data. Contributions include the new attack scenario and how to carry out the attack in a realistic setting with minimal assumptions on the attacker.
>
> > give little indication of broader implications.
>
> Breadth of implications:The main implication is that practitioners should be very careful when releasing models that have been trained on overlapping datasets since the difference between the datasets, as we show, can be leaked. Since releasing updated models is common (e.g., due to GDPR), we believe it is a serious concern the language modelling community needs to be aware of.
>
> > It would have been more convincing if these results replicated with two
> > splits of the same dataset, rather than identical datasets with one
> > augmented by canary tokens.
>
> Since the submission deadline, we performed a range of experiments to  evaluate information leakage in other data-overlapping scenarios. The updated submission presents these results in Table 2 and in Section 3.2. Summary of this experiment is given in the general response above.
>
> > The qualitative evaluation of subject-specific updates is also not
> > sufficiently informative. It would have been useful to define a
> > specific attack and see under what circumstances such an attack would
> > succeed. In the current results, I am not convinced that any of the
> > phrases in Table 3 represent a privacy violation.
>
> The qualitative evaluation in Section 3.3 shows that our attacks recover phrases related to the content of the data used to update the model rather than to the rest of the data. If this data has been selected from private conversations instead of public discussions in a newsgroup, an attacker would be able to infer recurrent conversation topics, violating the privacy of the participants.
>
> To define what it means for a specific attack to succeed, we would need a quantitative measure of success. We are exploring one such measure: train a classifier that discriminates between the public and training data used to update a model and compute the sum of the probabilities with which the discriminator classifies the phrases extracted as belonging to the private data. We think that the phrases output by our attack would be overwhelmingly classified as coming from the private data. We expect to include the results of our experiments when finalizing the submission.
>
> > The differential privacy experiment seems to be missing many details:
>
> We have updated Section 4 as per reviewer’s comments and outline them below:
>
> > what dataset was this trained on?
>
> We used the Penn Treebank dataset.
>
> > Are the accuracy values for the training set or a separate testing set?
>
> All accuracies that we report are for a separate validation set.
> -	Training accuracies are (29.73%,11.52%, 13%) for (non-DP, eps=5, eps=111), respectively.
> -	Validation accuracies are (23%, 11.89%, 13.34%) for (non-DP, eps=5, eps=111), respectively.
>
> We note that the discrepancy between training and validation accuracies is consistent with previous results on DP training.
>
> > Other works have shown that it is possible to train a differentially
> > private language model without large sacrifices in accuracy, so it
> > would be helpful to know what differentiates this experiment.
>
> The language models trained in McMahan et al. consider user-level privacy (i.e., batch of token-sequences), while we consider single token-sequence-level privacy. Hence, the gradient clipping is done per sequence in our case and not per batch. The difference between batch-level data in terms of gradients is smaller than that of sequence-level gradients (intuitively the differences are ``averaged’’ in a batch). As a result, updates are not “too different” between users. Hence, the noise, which is proportional to the change in the gradient, that needs to be added is much smaller when guaranteeing user-level privacy as compared to sequence-level privacy.
> The model trained by Carlini et al. is for a character prediction task which is a much simpler task than token prediction considered in our paper.
> That said, training privacy-preserving models with sequence-level privacy and good utility is an important research question but out of scope for this work.

---

> ### Author Response · Authors · 2019-11-08
> **Discussion of Review #3 (Part 2/2)**
>
> > I would also note that the motivation, a predictive keyboard, is not a
> > situation in which maximizing accuracy is generally desirable: users
> > tend to find this creepy rather than helpful. This is a nice idea but
> > would benefit from some more polishing and more extensive testing.
>
> In the Smart Compose setting, the user, as she is typing her email, is given several choices for the next token. In order for Smart Compose to be useful, there should be an intersection between what the user intends to write and the choices suggested. Hence, maximizing accuracy is important, though of course striking a balance to avoid “too personal” suggestions is important. How to strike this balance is out of scope for this paper.
> However, if models are personalized (or at least customized to a group similar to a specific user), they do have to shift their recommendations slightly to better match the data distribution of the data used for customization. In our submission we argue that this _shift_ is already leaking private information. You seem to be referring to the “My social security number is” prefix setting of the Secret Sharer work of Carlini et al. (2019), in which the leakage happens because private information is the most likely prediction. However, our analysis of model updates shows that leakage also happens when (a) the leaked data is _not_ the top prediction of any individual model and (b) no prefixes are available.

---

### Official Review · AnonReviewer2 · 2019-10-23
**Official Blind Review #2**

**Rating:** 3

**Review:**

This paper studies the privacy issue of widely used neural language models in the current literature. The authors consider the privacy implication phenomena of two model snapshots before and after an update. The updating setting considered in this paper is kind of interesting. However, the contribution of the current paper is not strong enough and there are many unclear experimental settings in the current paper.

According to the current paper, the privacy implication seems to be defined in terms of general sequences in training datasets. If this is the case, I don’t think such privacy implication is meaningful because our language models should memorize some general information to achieve their tasks.

There are some unclear settings in the experiments:
1.In the experiments, why there are only 20000 vocabulary size for Wikitext-103 datasets?
2.It is unclear how to construct canary phrases.
3.After constructing the new dataset, the model is retrained or trained in the online way?
4.Since the results for Wikitext-103 is not finished, the authors should remove the results on this dataset.
5.What is the perplexity of the trained models?
6.How to choose initial sequence in real data experiments?
7.When you applying DP mechanism, how did you define the neighboring datasets, and how did you implement it (what is the clipping level, how did you calculate privacy loss for language models)?
8.$\epsilon=111$ seems that the model will provide no privacy guarantee according to the definition of differential privacy?

**Experience Assessment:**

I have read many papers in this area.

**Review Assessment: Checking Correctness Of Derivations And Theory:**

I carefully checked the derivations and theory.

**Review Assessment: Checking Correctness Of Experiments:**

I carefully checked the experiments.

**Review Assessment: Thoroughness In Paper Reading:**

I read the paper thoroughly.

---

> ### Author Response · Authors · 2019-11-08
> **Discussion of Review #2**
>
> Thank you for engaging with our submission and asking questions!
>
> > According to the current paper, the privacy implication seems to be
> > defined in terms of general sequences in training datasets. If this is
> > the case, I don’t think such privacy implication is meaningful because
> > our language models should memorize some general information to
> > achieve their tasks.
>
> Our experiments show that much more than general information is revealed through model updates, because _specific_ phrases occurring in the training data as rarely as one in a million times can be extracted.
> As an example, consider the case of a “Smart Compose” feature (i.e., email auto-completion) using a model trained on data from a given company, and from which employees can extract phrases of the form “We will close [city] office”, because similar phrases occur multiple times in emails of C-suite managers.
>
> > 1. In the experiments, why there are only 20000 vocabulary size for
> > Wikitext-103 datasets?
>
> 20k was primarily chosen for performance reasons, both during our experiments as well as when considering the application scenario of predictive keyboards on client devices (where deploying the full Wikitext-103 vocabulary of size 267k would be infeasible in most cases). While 20k is indeed somewhat arbitrary, we are confident that increasing the vocabulary size (i.e., increasing the capacity of the model) would not change the direction of our results in a substantial way.
>
> > 2. It is unclear how to construct canary phrases.
>
> Our experiments use canaries that
>  * serve as a proxy for "private data", i.e. they should be grammatically correct but must not appear in the original dataset, and
>  * exhibit different token frequency characteristics
> To this end, we choose different valid phrase structures (e.g. Subject, Verb, Adverb, Compound Object) and instantiate each placeholder with a token that has the desired frequency characteristic in the dataset under consideration.
> For our experiments we construct the canaries manually, but automation is straightforward. We updated the description of the canary construction in Section 3.2. accordingly.
>
> > 3. After constructing the new dataset, the model is retrained or
> > trained in the online way?
>
> The submitted version of our paper retrained the model from scratch. However, we have updated the paper with experiments using a continued training setup, in which the model is trained on the original data, and then “fine-tuned” on a smaller additional dataset. Please see RQ3B and Table 2 in Section 3.2 and its summary in the general response.
>
> > 4. Since the results for Wikitext-103 is not finished, the authors
> > should remove the results on this dataset.
>
> We have updated Table 1 in the paper with the results for Wikitext-103.
>
> > 5. What is the perplexity of the trained models?
>
> The validation perplexity for the trained models is as follows (we added this information to Table 1):
> Penn-Treebank: 120.90
> Reddit (RNN): 79.63
> Reddit (Transformer): 69.29
> Wikitext-103: 48.59
>
> > 6. How to choose initial sequence in real data experiments?
>
> When we compute the differential rank in RQ1,2,4-7, we compare the
> changes in probability of all token sequences [*], that is, there is no
> need for the adversary to choose any specific initial sequence.
> In RQ3 we show that partial knowledge (i.e., knowledge of an initial
> sequence) about data used in the update can lead to more effective attacks.
>
> [*] In practice, we approximate this with a beam search, as discussed in Section 2.4.
>
> > 7. When you applying DP mechanism,how did you define the neighboring
> > datasets, and how did you implement it (what is the clipping level, how
> > did you calculate privacy loss for language models)?
>
> We use sequence-level differential privacy: i.e., two neighbouring datasets differ in a single sequence of tokens.
> We used the TensorFlow Privacy library for:
> (1)	Training with differentially private SGD. We used the Sampled Gaussian Mechanism that is provided by the library with the following parameters:
> - For eps=5: noise_multiplier=0.7, l2_norm_clip=5.0, sampling probability= .0048
> - For eps=111: noise_multiplier=0.3, l2_norm_clip=5.0, sampling_probability= .0024.
> (2)	Computing privacy loss. The library uses a Renyi differentially privacy accountant for computing the total privacy loss.
>
> > 8. \epsilon = 111 seems that the model will provide no privacy guarantee
> > according to the definition of differential privacy?
>
> Indeed, for a large epsilon, DP provides weak theoretical guarantees. However, our experiments show that it can still provide effective protection against our attack. This confirms results reported by Carlini et al. who show that current DP analyses come with (potentially overly) conservative bounds.

---

### Author Response · Authors · 2019-11-08
**General Review Response and Overview of Paper Revision**

We would like to clarify the core contributions of our paper. Training ML models on private data raises concerns about how much of this data is memorized and leaked by the models. In this paper, we advance the state-of-the art in this space as follows:

1.	We analyze privacy in an important novel attack scenario: Given API access to two models, one trained on a dataset $D$ and the other on $D + \Delta$,  where $\Delta$ includes private data, is it possible to extract information about $\Delta$? This question needs to be answered, for example, when augmenting models that are pre-trained on massive public datasets with private data, and when deleting a user’s data from a dataset, e.g., following GDPR.

2.	We show that the threat to privacy is real: An attacker can successfully recover information about $\Delta$ from the inference outputs of the models. The attack is effective even without background knowledge about $D$ or $\Delta$.


= Summary of changes we made to the paper:
1. Added missing values for Wikitext-103 model in Table 1.
2. Added the validation perplexity of each of the models in Table 1.
3. Added experimental results on re-training with different data splits and continued training. RQ3A and RQ3B and Table 2 summarize the results of our experiments.
4. Clarified the construction of canaries in Section 3.2.
5. Clarified DP experiments in Section 4.

= Summary of new experimental results (Table 2, Section 3.2):
The original submission extracted information about $\Delta$ between model $M$, trained on data $D_{orig}$, and updated model $M’$, trained on $D_{orig}$ and $\Delta$, where $\Delta$ was either canary phrases or a newsgroup. We added results for the following settings:
1.	$M’$ is trained on $D_{orig}$ +canaries+$D_{extra}$ [RQ3A in Section 3.2]:: Additional text $ D_{extra}$ did not affect the differential score (DS) of the canary phrase as the score remained constant for different splits between $D_{orig}$ and $D_{extra}$. Thus, the canaries are susceptible to leakage even when the updated model is trained using additional dataset.
2.	Continued training [RQ3B in Section 3.2]: $M’$ is initialized with parameters of $M$ and is trained further with new data $D_{extra}$ and canaries. In this setting, we observed that DS values have increased (i.e., higher susceptibility to leakage) in comparison to when $M’$ is trained from scratch on $D_{orig}$+$D_{extra}$+canaries.
3.	Continued training with two stages [RQ3B in Section 3.2]: An intermediate model $\tilde{M}$ is updated as above, i.e., initialized with $M$ and updated with $D_{extra}$+canaries. The final model, $M’$ is trained starting from $\tilde {M}$ and extra dataset $D’_{extra}$. We observed that the DS substantially reduces in this setting making it suitable as a potential mitigation strategy.

---

> ### Author Response · Authors · 2019-11-15
> **Addendum**
>
> The latest revision adds experimental results on re-training with different data splits and continued training on real-world data from the 20 Newsgroups dataset. See RQ4A, RQ4B and Table 5 in Section 3.3 for an analysis of the results.
> This expands the experiments on synthetic data (canary phrases) added in the previous revision.

---

### Decision · Program_Chairs · 2019-12-19

**Decision:**

Reject

**Comment:**

This paper report empirical implications of privacy ‘leaks’ in language models. Reviewers generally agree that the results look promising and interesting, but the paper isn’t fully developed yet. A few pointed out that framing the paper better to better indicate broader implications of the observed symptoms would greatly improve the paper. Another pointed out better placing this work in the context of other related work. Overall, this paper could use another cycle of polishing/enhancing the results.